# Navigating Tumour Microenvironment and Wnt Signalling Crosstalk: Implications for Advanced Cancer Therapeutics

**DOI:** 10.3390/cancers15245847

**Published:** 2023-12-14

**Authors:** Shraddha Shravani Peri, Krithicaa Narayanaa Y, Therese Deebiga Hubert, Roshini Rajaraman, Frank Arfuso, Sandhya Sundaram, B. Archana, Sudha Warrier, Arun Dharmarajan, Lakshmi R. Perumalsamy

**Affiliations:** 1Department of Biomedical Sciences, Faculty of Biomedical Sciences, Technology and Research, Sri Ramachandra Institute of Higher Education and Research, Chennai 600116, India; shraddhaperi@gmail.com (S.S.P.); krithicaanarayanaay@sriramachandra.edu.in (K.N.Y.); therese.deebiga@gmail.com (T.D.H.); roshinirajaraman65@gmail.com (R.R.); 2School of Human Sciences, The University of Western Australia, Nedlands, WA 6009, Australia; frank.arfuso@uwa.edu.au; 3Department of Pathology, Sri Ramachandra Institute of Higher Education and Research, Chennai 600116, India; sandhyas@sriramachandra.edu.in (S.S.); archanab@sriramachandra.edu.in (B.A.); 4Department of Biotechnology, Sri Ramachandra Institute of Higher Education and Research, Chennai 600116, India; sudha.warrier@sriramachandra.edu.in; 5Curtin Health Innovation Research Institute, Curtin University, Perth, WA 6102, Australia; 6Curtin Medical School, Curtin University, Perth, WA 6102, Australia

**Keywords:** Wnt signalling, tumour microenvironment, tumour-infiltrating lymphocytes, cancer-associated fibroblasts, cancer-associated adipocytes, tumour-activated macrophages, tumour vasculature, cancer stem cells, Wnt inhibitors, cancer therapy

## Abstract

**Simple Summary:**

The tumour microenvironment comprises cellular components and the extracellular matrix, which surround the core of cancer cells and play a vital role in their development. Among these components are various immune cells such as T-regulatory cells, dendritic cells, and tumour-infiltrating lymphocytes, as well as other cells such as adipocytes, endothelial cells, cancer-associated fibroblasts (CAFs), cancer stem cells, and neuroendocrine cells, among others. The Wnt signalling pathway is a central pathway involved in tumour development. This review explores the significance of Wnt signalling in the communication between cellular components within the tumour microenvironment. Understanding this signalling pathway opens up possibilities for modifying the tumour microenvironment, potentially leading to insights into tumour development and progression. Additionally, this review delves into Wnt inhibition and inhibitors and their potential application in cancer therapeutics.

**Abstract:**

Cancer therapeutics face significant challenges due to drug resistance and tumour recurrence. The tumour microenvironment (TME) is a crucial contributor and essential hallmark of cancer. It encompasses various components surrounding the tumour, including intercellular elements, immune system cells, the vascular system, stem cells, and extracellular matrices, all of which play critical roles in tumour progression, epithelial–mesenchymal transition, metastasis, drug resistance, and relapse. These components interact with multiple signalling pathways, positively or negatively influencing cell growth. Abnormal regulation of the Wnt signalling pathway has been observed in tumorigenesis and contributes to tumour growth. A comprehensive understanding and characterisation of how different cells within the TME communicate through signalling pathways is vital. This review aims to explore the intricate and dynamic interactions, expressions, and alterations of TME components and the Wnt signalling pathway, offering valuable insights into the development of therapeutic applications.

## 1. Introduction

Cancer, the sixth leading cause of fatality worldwide, is characterised by the uncontrolled growth of cells, resistance to apoptosis, and the ability to evade the body’s natural regulatory mechanisms, enabling relentless growth and survival against the body’s normal constraints. The revised hallmarks of cancer proposed by Douglas Hanahan include the features and capabilities acquired by the cancer cells that can be referred to [1]. Aberrant Wnt signalling has emerged as a critical mechanism in cancer biology and is known to be altered in various cancer types. The Wnt pathway is pivotal in regulating the proliferation and differentiation of stem cells and progenitor cells from embryogenesis to adult tissue homeostasis. It can be classified into canonical (β-catenin-dependent) and noncanonical (β-catenin-independent) pathways, which include the calcium pathway and planar cell polarity pathway (PCP) [2]. The Wnt signalling cascade is activated by Wnt ligands interacting with frizzled receptors, leading to diverse downstream effects [3].

The dysregulation of Wnt signalling in cancer impacts crucial tumour properties such as proliferation, invasion, migration, epithelial–mesenchymal transition, metastasis, apoptosis, senescence, therapy resistance, immune evasion, and relapse. Altered Wnt signalling has been observed in various cancer types, such as colon, breast, leukaemia, and melanoma. This pathway is influenced by a myriad of intracellular and extracellular components, endogenous and exogenous inhibitors, and cross-talk with other pathways, further contributing to cancer progression [4,5]. Figure 1 and Figure 2 illustrate the canonical and noncanonical Wnt signalling pathways.

The tumour microenvironment (TME) surrounding a tumour consists of various components, including blood vessels, fibroblasts, immune cells, signalling molecules, hormones, cytokines, and the extracellular matrix. Dysregulation of cellular signalling within the TME is recognised as a critical contributor to cancer development and progression [6]. Wnt signalling has been found to interact with different components of the TME in context-specific mechanisms. Wnt signalling promotes the communication between tumour cells and the TME, which is essential for tumour development and progression. Stromal cells and adjacent epithelial cells’ equilibrium are vital for tissue homeostasis, and any disruptions can lead to tumorigenesis. TME components, such as tumour-infiltrating immune cells, can create a pro-tumorigenic microenvironment by secretion of inflammatory markers [7,8,9,10,11,12]. Additionally, Wnt signalling can bridge tumour cells and the TME to influence tumour metastasis, invasion, and tolerance [13].

This review aims to summarise the intricate role of Wnt signalling in regulating the cellular components of the TME. It will explore the interplay between Wnt signalling and the TME (summarized in Figure 3), providing insights into potential therapeutic strategies targeting the Wnt pathway for cancer treatment.

## 2. Wnt-Mediated Immune Editing

The immune system prevents the transformation of healthy cells into tumour cells. However, under certain conditions, the immune system may exhibit tumour-promoting activity instead of its protective function. This phenomenon is known as immune editing, where enduring attacks from immune cells induce cancer [14]. Various immune cells, such as B-cells, T-cells, natural killer (NK), dendritic cells (DC), macrophages, cytokines, and neutrophils, are pivotal in developing anticancer responses by eliminating cancer cells. Cytotoxic T-cells (CD8^+^) are critical in immune editing, as they recognise and eliminate cancer-specific antigens, inducing apoptosis in cancer cells. Alternatively, regulatory T-cells (Tregs) play a suppressive role in the immune system, maintaining immune homeostasis and preventing excessive immune responses. However, Tregs can hinder anti-tumour immune responses in the cancer environment by suppressing cytotoxic T-cells and other immune cells and promoting immune evasion by cancer cells [15]. Immune evasion is a strategy employed by cancer cells to escape immune surveillance and attack. Myeloid-derived suppressor cells (MDSCs) are immature myeloid cells that accumulate in the TME and inhibit the function of cytotoxic T-cells and NK cells, creating an immunosuppressive microenvironment that supports tumour growth and immune evasion [16]. Macrophages can exhibit different phenotypes depending on their activation state, with M1 macrophages having anti-tumour properties and M2 macrophages having immunosuppressive properties [17].

Cytokines act as messengers that modulate the immune response to injuries and infections, mediating interactions within the TME and Wnt pathway and influencing various aspects of tumorigenesis. Cytokines alter the behaviour of tumour cells by regulating the self-renewal, survival, and drug resistance of cancer stem cells by interacting with several signalling pathways [18,19]. Toll-like receptor signalling gets activated after recognising pathogens and induces inflammatory-related responses. The Wnt ligands in the TME could interact and activate the TLR pathways. These TLR signalling mechanisms enhance the secretion of cytokines [20]. Inflammatory cytokines, such as interleukins and tumour necrosis factor (TNF), can create a pro-inflammatory environment that supports cancer development. The noncanonical Wnt pathway interacts with Toll-like receptors (TLRs) in macrophage cells. The pro-inflammatory cytokine, Interleukin 1β (IL-1β), phosphorylates and inactivates GSK3β, enhancing Wnt signalling mediated transcription of β-catenin target genes in colon cancer [21]. 

Wnt signalling operates contextually to modulate immune responses across diverse conditions. Within the TME, where immune cells are abundant, Wnt signalling yields beneficial and detrimental effects on tumour evolution and progression [13]. Numerous Wnt proteins activate immune responses in effector cells during the early stages of cancer development, acting as tumour suppressors; however, they can foster immune tolerance in the metastatic phase, enabling evasion from immune control [22]. Wnt signalling governs B-cell differentiation, thymocyte development, thymic selection, thymic degeneration, and immune tolerance, underscoring its significance within the TME [23]. Wnt signalling plays a pivotal role by halting T-cell differentiation and fostering the generation of memory stem cells, contributing to long-term immunological memory [24]. In hepatocellular carcinoma, Wnt/β-catenin signalling contributes to tumour growth, migration, metastasis, and immunosuppression [25]. In colon cancer, β-catenin activation triggers the production of T helper 17 (TH17) cell-mediated inflammation that propels cancer function. Similarly, in non-small-cell lung cancer, the FOXP3 gene, a master regulator of Treg cells, facilitates enhanced β-catenin-TCF-4 complex formation, leading to increased epithelial–mesenchymal transition (EMT), tumour growth, and metastasis [26]. Wnt/β-catenin signalling curbs the immunosuppressive actions of these cells by modulating TCF-1-dependent inhibition of Foxp3 transcriptional activity. Conversely, disruption of Wnt activity boosts the suppressive potential of Treg cells [15,27,28]. Conversely, overexpression of β-catenin in melanoma thwarts tumour immune evasion by restraining T-cell infiltration and subduing immune cell function [29]. In colon cancer, elevated β-catenin expression is observed in T-cells and Tregs of patients with colitis and colon cancer [30]. A recent study in pancreatic ductal adenocarcinoma identified Wnt targets in T-cells and the potential regulation of immunosuppressive tumour microenvironment through scRNA-sequencing. Disruption of TCF7, a transcriptional mediator in the WNT pathway, within CD4^+^ T-cells and the pharmacological suppression of ligand-mediated WNT signalling leads to increased infiltration of CD8^+^ T-cells and suppresses the growth of pancreatic cancer. This highlights Wnt signalling as a crucial regulator influencing the immune microenvironment in pancreatic cancer [31].

Tumour-infiltrating lymphocytes (TILs) encompass CD4^+^ T-cells, CD8^+^ T-cells, and NK T-cells, which infiltrate the tumour site from the bloodstream. Wnt signalling is intricately involved in the normal differentiation and functioning of CD4^+^ and CD8^+^ T-cells. TILs are recognised to heighten Wnt activity within cancer cells, fostering tumour progression and development. Notably, in breast cancer, particularly HER2-enriched and triple-negative breast cancer, TILs induce overexpression of β-catenin, holding potential as a regulatory target for immune infiltrates [32]. The canonical Wnt signalling inhibitor Dickkopf (DKK)1 is overexpressed in various TMEs and serum samples of patients with pancreatic, gastric, hepatic, bile duct, breast, and cervical carcinomas, eliciting host-protective immunity [33]. In a murine melanoma model, DKK1 vaccination incited protective immunity mediated by CD4^+^ and CD8^+^ T-cells [34]. Successful recognition of tumour-associated antigens and consequent eradication of cancer by cytotoxic T lymphocytes (CTLs) can be hindered by checkpoint proteins such as cytotoxic T lymphocyte-associated protein 4 (CTLA-4) or the programmed cell death 1 (PD-1) and programmed death-ligand 1 (PD-L1) pathways. GSK-3, a pivotal Wnt pathway regulator, can dampen CD8^+^ T-cell function by elevating PD-1 expression [35]. Disrupting interactions between Wnt and its cognate co-receptor LRP5/6 and frizzled has indicated that LRP5 and LRP6 expression in dendritic cells (DCs) plays a pivotal role in immune tolerance [36]. Moreover, β-catenin/mTOR/IL-10 signalling impedes DCs’ ability to cross-prime CD8^+^ T-cell immunity [37]. 

Recent findings suggest that WNT5a released from melanoma cells prompts paracrine WNT5-β-catenin signalling in DCs, resulting in an upsurge of the immunoregulatory enzyme indoleamine 2,3-dioxygenase-1 (IDO), which is pivotal in tumour-induced immune tolerance [38]. Tumour-cell-derived Wnt ligands drive M2-like polarisation of tumour-associated macrophages (TAMs) via the canonical pathway [38]. In colon cancer, interleukin-1 (IL-1) and TAMs collaborate to activate NF-κB-dependent PDK1/AKT signalling in tumour cells, inactivating glycogen synthase kinase 3 beta (GSK3β), amplifying Wnt signalling, and fostering colon cancer cell proliferation [39]. In cancer, upon exposure to microbial challenges, the activation of dectin-1, a receptor in macrophages, stabilises β-catenin and stimulates the increased secretion of noncanonical Wnt5a [40]. Subsequently, in myeloid cells, Wnt5a binds to specific TLRs (such as TLR2/4) [41], playing a distinctive role in downregulating the expression of proinflammatory cytokines like IL-12, IL1b, and TNF-a. This interaction induces immune tolerance in cancer cells [42,43], showcasing a unique mechanism by which the Wnt signalling pathway modulates the inflammatory response in the context of microbial challenges. This unveils a novel molecular connection between inflammation and tumour growth. In liver cancer, lipid accumulation triggers steatosis, and macrophage infiltration exacerbates Wnt expression, activating Wnt/β-catenin signalling. This fosters the proliferation of tumour-initiating cells and instigates tumorigenesis, thereby heightening liver cancer risk in obese individuals [44]. A comprehensive compendium of Wnt-mediated immune responses in the TME and corresponding immuno-therapeutic approaches is expounded upon in the review by Patel and colleagues [15].

Balancing the activities of these immune cells within the TME can critically influence the outcomes of immune editing. While the intricacies of interactions between Wnt signalling and immune editing are yet to be fully elucidated, these findings underscore the potential involvement of Wnt signalling in shaping the TME and impacting immune responses against cancer. Strategies that bolster cytotoxic immune cell efficacy while dampening immunosuppressive cell actions, such as Tregs and MDSCs, hold promise as potential immunotherapies for cancer treatment.

### Wnt Signalling in Cancer Immunotherapies

The influence of Wnt signalling on immunotherapies, particularly its impact on sensitivity or resistance, is a complex area of investigation. Canonical Wnt/β-catenin pathway activation has been associated with immune evasion mechanisms in the tumour microenvironment, potentially leading to resistance to immunotherapies [45,46]. In colon cancer, hyperactivation of Wnt/β-catenin signalling could alter TAMs in the TME, protecting the tumour cells undergoing apoptosis and leading to a pro-tumoural phenotype, which could resist immune checkpoint inhibitors [47]. Amplification of immune checkpoint molecules, such as PD-L1, can inhibit T-cell function and contribute to immune suppression within the tumour [48,49]. On the other hand, noncanonical Wnt signalling, specifically Wnt5a, has demonstrated contrasting effects. Wnt5a activation has been linked to increased inflammation and immune cell infiltration, suggesting a potential role in enhancing sensitivity to immunotherapies. Studies indicate that Wnt5a may promote anti-tumour immune responses by influencing the composition and function of immune cells within the tumour microenvironment [50,51]. Wnt signalling fosters immune suppression within the TME by interacting with other cell signalling transductions, contributing to an immunosuppressive milieu that hampers effective antitumor immune responses [52,53].

Chimeric Antigen Receptor T-cell (CAR T-cell) therapy genetically modifies patient T-cells to express a chimeric antigen receptor (CAR) to target and eliminate cancer cells precisely. While initially designed for haematologic cancers, its application to solid tumours faces challenges due to shared antigens with normal tissues and the immunosuppressive tumour microenvironment [54,55,56]. Strategies include combining CAR T-cells with agents that modulate the TME, such as immune checkpoint inhibitors, Wnt inhibitors or cytokines [57]. The use of CAR T-cells in combination with immune checkpoint inhibitors or Wnt inhibitors aims to overcome inhibitory signals within the TME, allowing CAR T-cells to exert their cytotoxic effects more effectively. While the direct combination of CAR T-cell therapy and Wnt inhibitors is an area of ongoing research, a study in colon cancer found the expression of cancer antigen EpCAM and activation of Wnt signalling. Combining CAR T-cells targeting EpCAM with the Wnt inhibitor hsBCL9_CT_-24 promotes T-cell infiltration and inhibits tumour growth [58]. 

## 3. Wnt-Signalling-Driven Tumour–CAF Interaction in the TME

Fibroblasts, a type of stromal cell, play a vital role in regulating differentiation, homeostasis, and immune response. Within the TME, specialised activated fibroblasts known as CAFs significantly contribute to tumorigenesis, EMT, metastasis, and other processes. CAFs achieve these effects through the secretion of growth factors such as transforming growth factor-β, hepatocyte growth factors, fibroblast secreting protein-1, stromal-derived factor-1, and angiopoietin, and by modifying the extracellular matrix. Additionally, they utilise extracellular degrading proteins such as matrix metalloproteinases (MMPs) [8]. CAFs promote tumour evasion, immunosuppression, and neo-angiogenesis, further fuelling tumour progression [12,59].

The influence of Wnt signalling on CAFs within the TME is well-established. In colorectal carcinomas, the canonical Wnt pathway is activated in CAFs upon binding the Wnt2 ligand to the FZD8 receptor. This activation contributes to the tumour’s aggressive behaviour and poor prognosis [60,61]. Upregulation of specific genes expressing Wnt ligands, such as Secreted frizzled-related protein 2 (SFRP2), Wnt2, Wnt5a, Wnt7a, or Wnt-1-induced secreted protein 1 (Wisp-1), leads to the transformation of normal fibroblasts into CAFs within the tumour stroma in various cancers, including colorectal, gastric, breast, and head and neck carcinomas. This transformation is associated with increased invasiveness, stemness of cancer stem cells, and tumour progression [62,63,64,65]. 

Extracellular vesicles released by CAFs, bound to Wnt11 secreted mainly by breast cancer cells, activate the PCP pathway, leading to enhanced cell protrusions and motility [66]. In prostate adenocarcinoma, WNT16B activation in fibroblasts through NF-κB was shown to induce EMT in neoplastic prostate epithelium via paracrine signalling, promoting cancer cell survival and recurrence post-cytotoxic therapy [67]. Wnt signalling activation in CAFs leads to nuclear transcription factor activation, resulting in tumour growth, matrix stiffness, angiogenesis, and cancer cell invasion, all contributing to decreased prognosis [68]. Wnt2 enhances colorectal carcinomas’ angiogenesis and extracellular matrix remodelling [69]. FZD5/NF-κB/ER signalling activated by the Wnt5a ligand targets FOSL2, a transcription factor of CAFs, promoting vascular endothelial growth factor (VEGF)-independent angiogenesis in breast cancer [70].

Interestingly, CAFs also regulate the secretion of Wnt proteins that induce tumour progression. CAFs in colorectal, endometrial, and oesophageal cancers contribute to the secretion of Wnt5a, Wnt1, Wnt10b, and Wnt2, further promoting tumour proliferation, migration and relapse by activating canonical/noncanonical Wnt pathways [71,72,73,74].

Collectively, Wnt-mediated activation of CAFs in the TME establishes a continuous paracrine signalling loop that fosters cancer progression and invasion. The interaction between Wnt-activated CAFs and CAF-promoting Wnt signalling within the TME is associated with a poor prognosis across various cancer types. Unravelling and targeting this intricate crosstalk and its interplay with the extracellular matrix and immune cells represent promising avenues for future research in cancer therapeutics.

## 4. Wnt Signalling in Tumour Adipocytes: Regulation and Impact

Adipocytes, the primary stromal fat cells within the TME, play a crucial role in tumour development and undergo dynamic changes. Chronic inflammation in adipose tissue can contribute to carcinogenesis. A complex interplay exists between cancer cells and cancer-associated adipocytes [75]. Wnt signalling is a critical mediator in regulating adipocyte differentiation from preadipocytes into mature adipocytes. Activation of the Wnt pathway often reduces adipocyte differentiation across multiple cancer types [76,77]. Components of the Wnt pathway, including Wnt10b and β-catenin, can suppress adipogenic transcription factors, leading to inhibited adipocyte differentiation and maintenance of an adipocyte pool within the TME [78].

In colorectal and hepatocellular carcinoma, the BAMBI feedback loop involving Wnt/β-catenin signalling inhibits adipogenesis [79]. Similarly, multiple myeloma exhibits decreased adipogenesis and heightened osteo-lineage development through Wnt signalling [80]. Modulators of the noncanonical WNT pathway, such as FrzB, SFRP2, R-spondin1, WNT5A, and WNT5B, are overexpressed in fully differentiated adipocytes, contributing to induced EMT and increased tumour aggressiveness [81,82]. In pancreatic cancer, Wnt signalling displays both adipogenic and anti-adipogenic effects [83].

Adipocytes within the TME secrete adipokines that impact tumour behaviour. The Wnt pathway regulates adipokine secretion, notably upregulating leptin expression and secretion. Leptin, associated with tumour growth, angiogenesis, and inflammation, modulates the adipokine profile of tumour adipocytes, influencing their paracrine signalling to nearby cancer and stromal cells. Adipocytes contribute to ovarian cancer metastasis and chemoresistance through adipokine release and lipid transfer [84]. The Wnt inhibitor SFRP5 acts as a novel adipokine, suppressing invasion and migration in breast cancer [85].

Cancer cells release Wnt ligands that activate the Wnt pathway in neighbouring adipocytes, inducing gene expression and phenotype changes. Wnt signalling within adipocytes promotes stemness, as demonstrated in breast cancer where adipocyte-secreted IL-6 and leptin activate breast cancer stem cells through Notch, Wnt, and Sex determining region Y-box 2/octamer binding transcription factor 4/Nanog signalling [86]. Physical stressors in mammary adenocarcinoma mechanically activate the Wnt pathway, causing adipocyte de-differentiation and self-renewal [87]. This interplay alters adipokine secretion, influencing the behaviour of adjacent cancer and stromal cells. Moreover, disrupted Wnt signalling within the TME affects lipid-related processes, potentially altering nutrient availability for neighbouring cancer cells. Targeting Wnt signalling components or downstream targets may hold therapeutic promise in addressing cancer-associated adipocytes and their contributions to tumour progression.

## 5. Wnt Signalling Mediated Modelling of Tumour Vasculature and Lymphangiogenesis

The vasculature within tumours exhibits morphological abnormalities, increased permeability, and susceptibility to structural damage, mainly attributed to the absence of pericytes for vessel support. Extracellular matrix (ECM) proteins are discerning markers, allowing differentiation between tumour vessels and their normal counterparts. Many of these proteins actively foster tumour angiogenesis [88]. Exploiting these distinctions becomes a strategic avenue for targeting tumour blood vessels to impede tumour growth. This approach holds promise as it enables the selective destruction of tumour blood vessels while minimally affecting normal resting vessels, emphasising the potential for precise intervention in cancer therapy [89]. A well-organized vascular network is pivotal for the progression of tumours, with endothelial cells (ECs) being essential players in regulating angiogenesis and metastasis. Within the TME, ECs exhibit heightened responsiveness to signals, facilitating the development of tumour-infiltrating blood vessels. In the initial stages, specialised ECs undergo a transformative process, shifting their phenotype towards mesenchymal cells through a complex mechanism known as an endothelial-mesenchymal transition [90]. Extensive research has identified a strong connection between endothelial cells and Wnt signalling [91]. This signalling pathway holds significance in governing both tumour vasculature and lymphangiogenesis, contributing to the formation of blood vessels and lymphatic vessels within the TME.

The Wnt pathway’s target genes typically trigger the activation of pro-angiogenic factors such as VEGF, stimulating the proliferation of endothelial cells and fostering vessel creation within tumours. This process enhances the expansion of the tumour vasculature, ultimately furnishing oxygen and nutrients to support tumour growth. The Wnt pathway also influences EC differentiation and sprouting during angiogenesis. This effect is achieved by upregulating the expression of specific genes, including platelet-derived growth factor (PDGF) and angiopoietin-2. The induction of Wnt pathway-driven angiogenesis involves a signalling cascade characterised by LRP6-mediated APC/Asef2/Cdc42 activation triggered by the Wnt antagonist DKK2. The expression of DKK2 is particularly pronounced during the morphogenesis of endothelial cells [92]. Beyond the conventional pathway, the planar cell polarity pathway also expresses VEGF [93]. Furthermore, in postnatal retinas and tumours, noncanonical Wnt5a, facilitated by the Wnt secretion factor, contributes to the activation of Wnt signalling. This activation promotes micro-vessel density and enhances EC survival and proliferation [94]. Intriguingly, the receptor for SFRP2, known as Fzd5 and belonging to the Wnt/Fzd family, has been found to mediate SFRP2-induced angiogenesis through the calcineurin/nuclear factor of activated T-cells cytoplasmic 3 (NFATc3) pathway in ECs [95]. In endometrial cancer cells, SOX17 suppresses the Wnt/β-catenin-EMT axis, inhibiting migration [96].

Wnt signalling activation has been observed in various cancer types, including colorectal cancer, where activators such as Wnt2, Wnt4, TGM2, and the long noncoding RNA GAS5 play a role [97]. In malignant glioma, heightened expression of Wnt7 in Olig2^+^ oligodendrocyte precursor-like cells facilitates the invasion of glioma cells into the vasculature, and blocking Wnt signalling suppresses this invasion while improving the response to temozolomide therapy [98]. Moreover, HIF-1α activated via the Wnt/β-catenin pathway promotes vasculogenesis and angiogenesis under normoxic conditions due to lactate released by glioma cells [99].

Wnt pathway activation expresses various angiogenic factors, including VEGF-C, TEM7, MMP2, MMP3, MMP9, PDF-1, HIF-1α, VEGFR2, IL-6, IL-8, COX2, and DMH, influencing angiogenesis across diverse tumour types [91,100,101]. In glioblastoma models, the Wnt signal’s activation through a c-Met-mediated axis, inducing phosphorylation of β-catenin at Ser675, heightens EC stemness and chemoresistance [102]. Additionally, Wnt signalling plays a role in recruiting pericytes, specialised cells that stabilise blood vessels, into the TME. These pericytes contribute to vessel maturation and integrity by secreting PDGF-BB and facilitating their interaction with ECs [103,104].

The Wnt pathway also extends its impact to lymphangiogenesis within the TME, influencing the differentiation and migration of lymphatic ECs. Wnt ligands interact with receptors on lymphatic ECs, prompting proliferation and sprouting to form new lymphatic vessels [105]. Considering the significance of lymph node metastases in tumours and their correlation with reduced patient survival, enhancing lymphatic vasculature development via noncanonical pathways by Wnt signalling presents a noteworthy finding [106]. Oscillatory shear stress intensifies canonical Wnt signalling in lymphatic endothelial cells (LECs), where lymphedema-associated transcription factors such as GATA2 and FOXC2 are expressed [107]. Moreover, the master transcription factor for LECs, PROX1, interacts with β-catenin and the TCF/LEF transcription factor TCF7L1, amplifying Wnt signalling and promoting lymphatic vascular development. Conversely, miR-10527-5p curbs metastasis and lymphangiogenesis by targeting Rab10-mediated canonical Wnt signalling in oesophageal squamous cell carcinoma [108]. The noncanonical Wnt pathway contributes to lymphangiogenesis via RAC and JNK pathways, fostering lymphatic network elongation, increased tube length, and horizontal migration in vivo [106].

Wnt signalling’s interplay with other pathways implicated in angiogenesis and lymphangiogenesis, such as VEGF and Notch pathways, further influences the formation and remodelling of tumour vasculature and lymphatic vessels within the TME. Understanding the intricate interactions between Wnt signalling and tumour angiogenesis and lymphangiogenesis within the TME is imperative for developing therapeutic strategies to control tumour vascularisation and lymphatic spread. 

## 6. Wnt Signalling in Neuroendocrine Cell Differentiation

Neuroendocrine cells specialise in producing and releasing hormones directly into the bloodstream, diverging from typical neurotransmitter release into synaptic clefts upon signals from neighbouring cells [109]. These cells, found across various organs, function as hybrids of neurons and endocrine cells. Within the context of cancer, the concepts of neuroendocrine tumours (NETs) and neuroendocrine cells within the TME are distinct. NETs arise in organs such as the gastrointestinal tract, pancreas, and lungs, where neuroendocrine cells are present. The expression of neuroendocrine markers, hormone secretion, potential for metastasis, and unique histological and molecular traits distinguish NETs [110]. Neuroendocrine cells in the TME refer to ordinary neuroendocrine cells present around tumours, found in proximity to or interspersed among cancer cells. These cells contribute to the microenvironment through paracrine signalling, hormone release, and intercellular communication [111]. Wnt signalling plays a pivotal role in preserving the fate of neuroendocrine cells by supporting the survival and proliferation of neuroendocrine progenitor cells while preventing their differentiation into other lineages. Notably, the FOXB2 transcription factor activates the WNT7B ligand in aggressive prostate cancer. Elevated WNT7B levels, when androgen signalling is absent, led to increased tumorigenesis and drug resistance [112]. Similarly, heightened Wnt11 levels influence the neuroendocrine differentiation of prostate cells, which is accompanied by increased mobility and resistance to apoptosis [113].

Wnt pathway interactions with other pathways within the TME, such as Notch, bone morphogenetic protein, and sonic hedgehog, are significant in shaping neuroendocrine differentiation. These interactions promote the commitment of progenitor cells to the endocrine lineage. Mutations in anaplastic lymphoma kinase (ALK) and c-myc activate the Wnt signalling pathway, driving neuroendocrine prostate cancer. Targeted inhibition of the ALK/c-myc/Wnt/β-catenin pathway could potentially hinder tumour progression [71]. Additionally, Wnt/β-catenin signalling regulates specific peptides, such as Neurotensin, contributing to elevated cell proliferation and anchorage-independent growth in various gastrointestinal cancers [114]. Specific genes, including ASCL1, regulate Wnt11 and other Wnt signalling molecules in small-cell lung cancer, initiating neuroendocrine differentiation and lung tumorigenesis [115]. Tumour-associated neuroendocrine differentiation can transpire in certain cancers such as small-cell lung carcinoma and NETs [116]. Aberrant Wnt signalling activation can confer neuroendocrine features upon non-neuroendocrine cancer cells, contributing to tumour heterogeneity and aggressiveness. Epigenetic mechanisms such as promoter methylation or histone modification of Wnt negative regulatory genes can lead to distorted Wnt pathways, as observed in pancreatic carcinoid tumours [117]. Similarly, mutations in β-catenin are implicated in gastric carcinoid tumours, driving alterations in the Wnt signalling pathway and initiating tumorigenesis in neuroendocrine cells [118,119].

Understanding the role of Wnt signalling in neuroendocrine cell differentiation within the TME and the emergence of NETs holds importance for deciphering the underlying mechanisms, thus paving the way for potential therapeutic strategies. Manipulating Wnt signalling or its interactions with other pathways presents a promising avenue for modulating neuroendocrine cell fate and effectively managing tumour growth and progression.

## 7. Wnt Signalling in Mesenchymal Stem Cell Regulation of TME

Mesenchymal stem cells (MSCs) constitute a group of multipotent stem cells originating from the mesoderm, showcasing self-renewal capabilities and the potential for differentiation into various cell types, including fibroblasts, osteoblasts, and adipocytes [120]. They play a pivotal role in maintaining the structure and function of connective tissue and produce specific bioactive molecules that facilitate tissue repair and regeneration [121]. In many cancers, their impact on tumour homing and immunomodulatory functions significantly influences the fate of tumours [120].

Wnt signalling plays a vital role in the recruitment and movement of MSCs, exercising control over their fate in the TME through the secretion of Wnt ligands by stromal or tumour cells [122]. This pathway guides the differentiation of MSCs into CAFs by reconfiguring extracellular matrix components such as collagen and fibronectin, releasing growth factors and cytokines via paracrine signalling, and influencing immune cell behaviour by attracting Tregs and MDSCs to create an immunosuppressive milieu [123]. These attributes collectively promote tumour cell attachment, advancement, migration, and invasion. In colon cancer, for instance, MSCs triggered downstream Wnt signals, generating factors such as FGF10, VEGFC, and MMPs associated with metastasis [124].

Notably, MSCs can act as regulators of the canonical Wnt signalling pathway, with overexpression of DKK-1 in MSCs acting as a significant antagonist in hepatocellular carcinomas [125,126]. This dual role of the Wnt pathway in MSCs extends to some cancers, wherein it operates as a tumour suppressor. For instance, in colon cancer, Wnt signalling ligands such as Wnt3a can stimulate adipose-derived MSCs to adopt a tumour-associated fibroblast-like phenotype associated with poor patient prognosis [127]. Additionally, the transformation of ECs into MSCs imparts therapeutic resistance in glioblastoma, and simultaneous targeting of temozolomide and Wnt/b-catenin-mediated endothelial-to-MSC transformation has shown potential in reducing glioblastoma tumour growth by overcoming chemotherapy resistance [102].

Comprehending the role of Wnt signalling in governing MSC behaviour within the TME is crucial for unravelling the intricate interplays between MSCs, tumour cells, and other stromal components. Targeting the Wnt pathway in MSCs or their secreted factors emerges as a prospective therapeutic strategy to manipulate the TME and enhance anti-tumour responses.

## 8. Wnt Signalling in Cancer Stem Cells Regulation of TME

Cancer stem cells (CSCs) constitute a small subset of tumour cells and possess the remarkable ability to self-renew, playing a critical role in tumour initiation, growth, and recurrence. These CSCs actively orchestrate the TME by engaging with components such as immune cells, CAFs, differentiated cells, blood vessels, and extracellular cues. Notably, the canonical Wnt pathway activation has been linked to heightened tumour progression and the preservation of CSC stemness and self-renewal [128]. Anomalies in Wnt signalling are responsible for metabolic process alterations such as glycolysis, glutaminolysis, and lipogenesis, which are vital for CSC survival and stemness. This contributes to tumour anti-immunity and resistance mechanisms [129].

A defining characteristic of CSCs is their ability to sustain extended telomeres through the upregulation of telomerase reverse transcriptase, with Wnt signalling playing a notable role in this context during tumorigenesis [130,131,132]. Activation of the Wnt pathway induces EMT transcription factors Snail, HIF-1, ZEB1, and STAT3, enhancing the metastatic potential of cancers, particularly those treated with ionising radiation where reactive oxygen species are implicated in Wnt activation [133]. Certain tumours exhibit the ubiquitination of core stem cell regulators (Nanog, Oct4, and Sox2) and components of the Wnt and Hippo-YAP signalling pathways to maintain CSC stemness [134]. The overexpression of *PER3* curbs sphere-forming capacities and dampens stemness in the colon and prostate cancer stem cells via the WNT/β-catenin pathway in the TME [135,136]. Wnt activation in breast cancer cells stimulates EMT and elevates mammosphere formation, accompanied by increased CSC markers and stem-like attributes [137]. In colon, lung, and skin cancers, factors such as AP4, RIF1, FOXC1, and hepatocyte growth factor drive Wnt signalling activation, fostering phenotypic changes such as maintaining CSC characteristics, metastatic potential, and cell cycle progression [138].

In gastric cancer, the interplay between gastric cancer cells and mesenchymal stem cells sustains the growth and properties of gastric CSCs in vivo, activating the R-spondin/Lgr5 axis and the WNT/β-catenin signalling pathway, as indicated by nuclear β-catenin localisation [139]. Conversely, the SFRP family of proteins, notably SFRP4, SFRP2, and SFRP1, serve as antagonists to Wnt signalling and have demonstrated efficacy in diminishing CSC traits such as drug resistance, stemness, metastasis, and proliferation across various cancer types. Moreover, they promote apoptosis and chemosensitisation [140,141,142,143,144,145].

## 9. Consequence of Wnt Inhibition on the TME

Wnt signalling is a crucial regulator of cancer progression across numerous cancer types. Several inhibitors have been developed and studied targeting different pathway steps. Figure 4 outlines the mechanisms through which Wnt signalling impacts the cellular elements of the TME. It is increasingly evident that Wnt ligand and receptor expression are crucial in the TME. Thus, Wnt inhibitors in the cancer environment will intercept the tumour cells and the TME. Several natural antagonists and synthesised chemical inhibitors of Wnt signalling are being explored for their role in regulating components of the TME (Figure 5). The role of Wnt inhibition in regulating tumour angiogenesis and interaction with CSCs is well characterised. Most inhibitors acting on ECs, such as Triptolide, GLA, miR-129-5p, miR-205-5p, (VEGFR)1/Flt1, work by inhibiting or negatively regulating angiogenesis [146,147,148,149,150,151]. γGPNA has been shown to cause a decrease in the expression of several pro-angiogenic secreted factors such as EphrinA1, FGF-2, and VEGF-A upon β-catenin inhibition in liver tumour cells [152]. CR-1 siRNA has been shown to suppress the secreted level of VEGF and cause a reduction in the protein level of VEGF-2. Another emerging aspect is the understanding of Wnt inhibition on CSCs. Most of the inhibitors work by reducing the property of stemness in CSCs or causing apoptosis of CSCs. Some inhibitors causing this effect include PRI-724, C59, Poziotinib, Salinomycin, miR-601; other Wnt inhibitors impact other TME components, such as adipocytes, TAMs, and CAFs [153,154]. These inhibitors target the β-catenin molecule and inhibit tumour progression, invasion, and metastasis. Table 1 summarises Wnt-TME-related therapeutic approaches. The comprehensive understanding of Wnt signalling in the context of the TME holds promise for developing effective cancer treatments. Various innovative agents are being investigated to alleviate the immunosuppressive environment within tumours by inhibiting the Wnt/β-catenin pathway. A combination of immune checkpoint and Wnt inhibitors (DKK1 and PORCN) are in clinical trials targeting various cancer models; for more on these, the reader can refer to [46,155]. Such studies suggest that inhibiting Wnt signalling may enhance the efficacy of immunotherapies.

## 10. Future Aspects and Conclusions

Evidence strongly indicates that Wnt signalling is a crucial link between tumour cells and the intricate TME, which collaboratively evolves to favour their selective growth and advancement. The TME comprises a heterogeneous mix of tumour cells, an array of growth factors, secreted molecules, and neighbouring stromal cells. Within this context, stromal cells and inflammatory cells situated in the extracellular matrix of the TME release Wnt ligands, fostering tumour invasion, metastasis, and tolerance. Through interactions involving diverse cellular constituents within the TME, Wnt signalling facilitates tumour progression. Furthermore, the TME’s various elements can modify Wnt signalling in tumour cells, thereby regulating the progression and metastasis of tumours in multiple contexts. Single-cell RNA sequencing, a high-throughput technique, is a powerful tool for studying the TME and intricate cellular responses to various signalling pathways. Understanding the single-cell responses to Wnt signalling can inform the development of targeted therapies [195,196]. The scRNA-seq data can reveal potential vulnerabilities or resistance mechanisms in specific cell populations, guiding the design of more effective therapeutic strategies. Critical inquiries are highlighted in Table 2, underscoring pivotal questions in this realm.

Wnt modulation has demonstrated its significance across a spectrum of cancer types. A range of inhibitors targeting Wnt signalling induces alterations or modifications within the TME. This, in turn, holds the potential for obstructing tumour progression, invasion, and metastasis. This perspective highlights how Wnt inhibitors could unleash therapeutic possibilities by targeting Wnt-mediated TME dynamics in tumours. In conjunction with established modalities such as surgery, chemotherapy, and radiotherapy, integrating these inhibitors to suppress Wnt signalling and its impact on TME components presents a promising avenue for adjuvant therapy, aiming to thwart and diminish cancer recurrence. Additionally, CAR T-cell therapy is an excellent therapeutic strategy combating the TME by precisely targeting cancer cells, overcoming challenges posed by the immunosuppressive TME and disrupting Wnt signalling pathways for enhanced antitumor efficacy.

Systematically charting the role of Wnt signalling on specific components within the TME for each cancer type is imperative for designing more effective therapeutic strategies targeting Wnt signalling in cancers. Emphasising the comprehension and management of the interplay between Wnt signalling and TME constituents could yield improved tumour diagnosis, prevention, and treatment avenues. Consequently, there is an urgent demand for a comprehensive characterisation of Wnt components, an in-depth understanding of Wnt signalling within the TME, and the associated causal consequences. Employing Wnt reporter models for in vitro studies and conducting in vivo analyses involving immunohistochemistry on primary and metastatic tumour patient tissue samples using specific antibodies to Wnt signalling components would yield valuable insights. Additionally, integrating single-cell analysis techniques in the TME domain would provide a deeper understanding of the heterogeneity of Wnt signalling’s role in TME regulation.

The foremost challenge in targeting the TME resides in the intricate interactions of neighbouring host cells and intercellular elements within this microenvironment. These complex and multifaceted interactions can harbour contrasting roles in regulating cancer cells. To tackle this complexity would require understanding their functions and interactions, deciphering the heterogeneous and dynamic attributes of these adjacent normal cells, and implementing suitable profiling and decoding strategies for the components of the TME, which would be instrumental in developing therapeutics targeting Wnt-mediated TME dynamics.

## Figures and Tables

**Figure 1 cancers-15-05847-f001:**
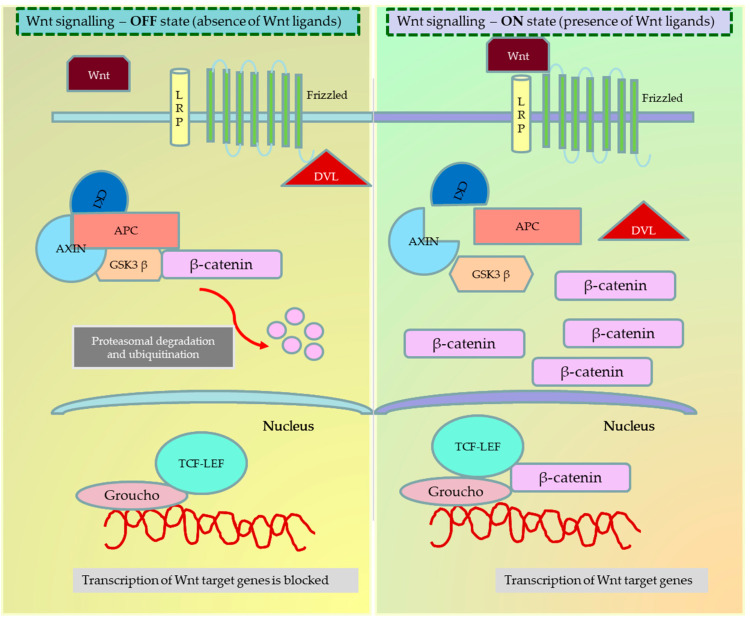
Wnt signalling—Canonical pathway in the ON and OFF state. When the frizzled family of receptors binds to the Wnt ligand, it recruits the dishevelled (Dsh) protein in the cell to frizzled receptors. Based on the dependency on β-catenin, the Wnt ligands are known to activate three different pathways—the canonical (β-catenin dependent) pathway, the noncanonical (β-catenin independent) pathway, and the noncanonical planar cell polarity pathway. In the ON state, the recruitment of Dsh to the cell membrane causes the disintegration of the APC degradation complex and the release of β-catenin. The free β-catenin can now translocate to the nucleus and bind to the T-cell factor/lymphoid enhancer-binding factor (TCF/LEF) family to activate the transcription of target genes. In the OFF state, the destruction complex phosphorylates and ubiquitinizes the β-catenin, preventing the translocation and activation of β-catenin target genes inside the nucleus. The figure was made in Microsoft Office PowerPoint.

**Figure 2 cancers-15-05847-f002:**
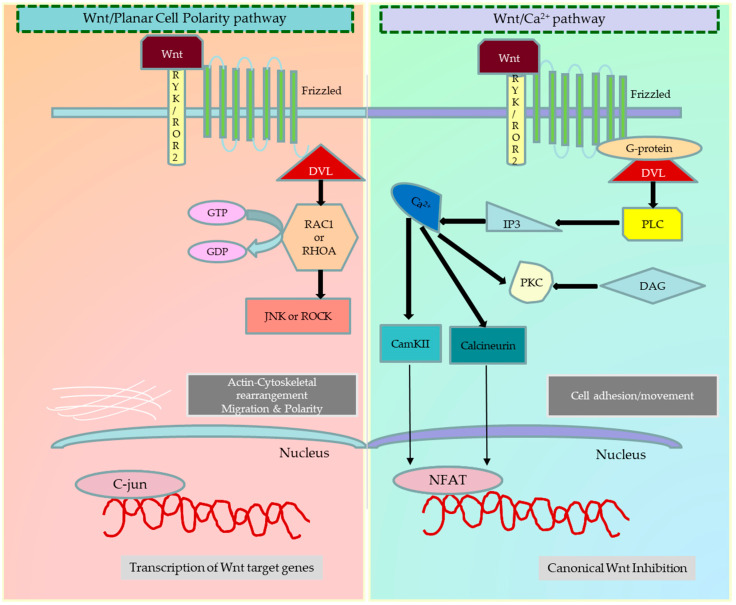
Different types of noncanonical Wnt signalling pathways. Activation of the planar cell polarity (PCP) pathway again depends on the association of (Fzd) protein with Dsh. The activated Fzd receptor recruits Dsh, which is associated with the dishevelled-associated activator of morphogenesis 1 (DAAM1), to activate the small G-protein Rho. Signalling downstream of Rho subsequently results in the modulation of the cytoskeleton. Dsh can also form a complex with Ras-related C3 botulinum toxin substrate 1 (Rac1), which regulates actin polymerisation. In addition, noncanonical Wnt signalling is also shown to use calcium as a secondary messenger to control cell fate. The presence of co-receptors in the proximity of Fzd receptors has been shown to influence the decision between canonical versus noncanonical Wnt signalling. The Wnt/Ca^2+^ pathway activates when the ligand binds the Fzd receptor and RYK/ROR co-receptors. This interaction increases the intracellular calcium levels and generates inositol 1,4,5-triphosphate-3 (IP3). The increased Ca^2+^ levels switch on the Ca^2+^ dependent calmodulin protein kinase (CaMKII), calcineurin, and protein kinase C (PKC). As a result, CAMKII and PKC phosphorylate the nuclear factor of activated T-cells (NFAT) and activate the transcription of target genes. The figure was made in Microsoft Office PowerPoint.

**Figure 3 cancers-15-05847-f003:**
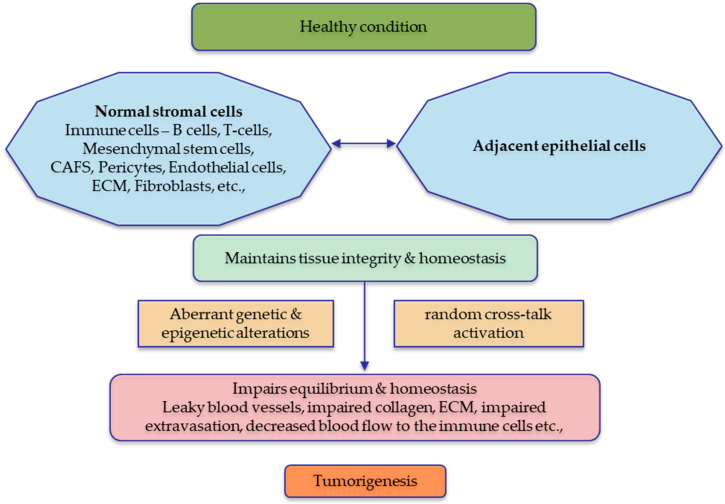
Tumour microenvironment in a healthy state vs. diseased state. The normal stromal cells are essential for maintaining the integrity of the neighbouring epithelial cells, and sustained cross-talk is observed for tissue homeostasis. Any aberrations in the normal stroma or the adjacent epithelial cells will significantly affect the stability of the tissues, which leads to tumorigenesis. The figure was made in Microsoft Office PowerPoint.

**Figure 4 cancers-15-05847-f004:**
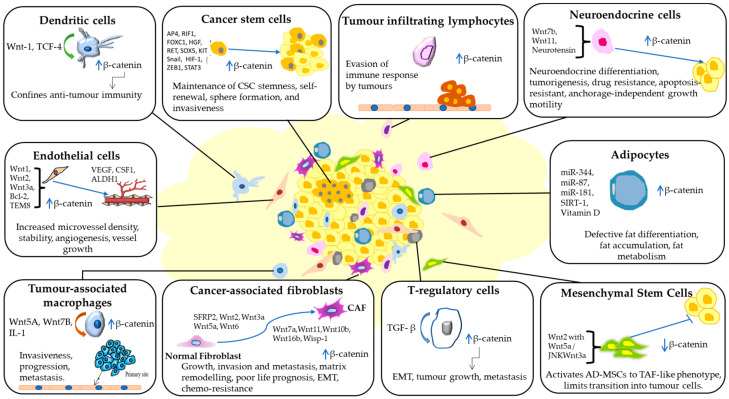
Wnt signalling orchestrates diverse interactions within the tumour microenvironment (TME). It drives the conversion of normal fibroblasts into cancer-associated fibroblasts (CAFs), stimulates tumour angiogenesis by endothelial cells (ECs), sustains stemness and self-renewal in cancer stem cells (CSCs), guides neuroendocrine and fat cell differentiation, and accelerates tumour progression, invasion, and metastasis. Moreover, Wnt signalling facilitates immune evasion by activating tumour-associated macrophages (TAMs), regulatory T-cells (Tregs), and tumour-infiltrating lymphocytes (TILs). The figure was made in Microsoft Office PowerPoint.

**Figure 5 cancers-15-05847-f005:**
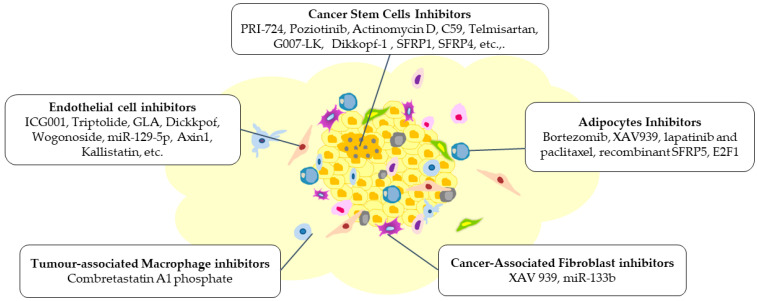
Modifications are brought about within the TME by various inhibitors targeting Wnt signalling, which can impede the tumour’s growth, invasion, and metastasis. These inhibitors have resulted in alterations of the different components of the TME, including cancer stem cells, endothelial cells, adipocytes, tumour-associated macrophages, and cancer-associated fibroblasts. The figure was made in Microsoft Office PowerPoint.

**Table 1 cancers-15-05847-t001:** Wnt Inhibitors in the TME.

Inhibitors	Target	Mode of Action	Conditions/Models	References
Combretastatin A1 phosphate	β-catenin	Microtubule inhibitor; TAM apoptosis.	Liver cancer	[146]
ICG001	CBP/β-catenin signal	Inhibits the proliferation and migration of endothelial cells.	Colon cancer	[149]
Triptolide	β-catenin	Inhibits angiogenesis.	Osteosarcoma	[139]
Glabridin (GLA)	Wnt-3′-untranslated regions (UTRs)	Inhibits angiogenesis.	Breast cancer	[140]
Dickkopf, tankyrases, and casein kinase 1 inhibitors	Fzd or LRP5/6	EC migration, sprouting and vascular tube formation.	Hodgkin lymphoma	[150]
Wogonoside	β-catenin	Inhibits angiogenesis.	Breast cancer	[151]
miR-129-5p	β-catenin	Inhibits angiogenesis.	Prostate cancer	[141]
Ginkgo biloba exocarp extracts	β-catenin	Inhibits angiogenesis.	Lung cancer	[152]
Diallyl trisulfide	TCF/LEF-mediated transcription; LRP6	Inhibits survival, migration, invasion, and angiogenesis in glioma cells.	Glioma	[153]
1118-20	β-catenin	Inhibits angiogenesis.	Liver cancer	[154]
P125A-endostatin and taxol	β-catenin	Inhibits angiogenesis.	Breast cancer	[155]
TNP470	Axin2, Myc and Ccnd1	Increased vessel regression accompanied by decreased EC proliferation.	Endothelial cells	[90]
Wnt inhibitory factor 1 and sFRP1	Wnt ligand-receptor interaction	Inhibits tube formation and migration of human microvascular endothelial cells and mouse endothelial progenitor cells (EPCs).	Liver cancer	[156]
Flt1	Wnt5a, Wnt11; VEGF	Inhibits angiogenic branching.	Myeloid cells	[143]
Kallistatin	Wnt3a	Inhibits angiogenesis, inflammation, tumour growth, and invasion in animal models and cultured cells.	Cancer	[157]
CR-1 siRNA	cyclin D1 and cyclin E1	Inhibits the secreted level of vascular endothelial growth factor.	Prostate cancer	[158]
B cell translocation gene 1	β-catenin	Inhibits angiogenesis.	Glioma	[159]
miR-205-5p	β-catenin	Inhibits angiogenesis.	psoriasis	[142]
sFRP2	Fzd receptor CRDs	Inhibits angiogenesis.	Angiosarcoma and Breast cancer	[160]
Cucurbitacin B	Wnt3 and Wnt3a	Inhibits angiogenesis.	Non-small-cell lung cancer	[161]
γ-tocotrienol	β-catenin, cyclin D1	Inhibits angiogenesis.	Gastric cancer	[162]
Phosphatidylinositol 3-kinase inhibitor	β-catenin	Inhibits angiogenesis.	Zebrafish embryos	[163]
(VEGFR)1/Flt1	Wnt5a, Wnt11, VEGF	Inhibits angiogenesis.	Colon cancer	[144]
XAV939	β-catenin	Inhibits angiogenesis and promotes apoptosis.	Lung cancer	[164]
gamma guanidine-based peptide nucleic acid	β-catenin	Decreases the expression of several pro-angiogenic secreted factors such as EphrinA1, FGF-2, and VEGF-A upon β-catenin inhibition.	Liver cancer	[145]
Tretinoin	β-catenin	Inhibits fibrogenesis and angiogenesis.	Liver cancer	[165]
XAV939	CAFs	Inhibits tumour cell proliferation and invasion.	Cervical cancer	[147]
miR-133b	CAFs	Targets the EZH2 gene and inhibits tumour proliferation, invasion, and migration.	Glioma	[166]
recombinant sFRP-5	Wnt5a–Fzd5 interaction	Blocks the dedifferentiation of adipocytes in co-culture.	Pancreatic cancer	[167]
lapatinib and paclitaxel	Wnt components through ErbB1 and ErbB2	Increases the expression levels of adipogenesis transcription factor peroxisome proliferator-activated receptor γ and Wnt antagonists (secreted frizzled-related protein 1, Dickkopf-related protein 1 and sclerostin) and inhibits angiogenesis.	Breast cancer and bone marrow damage	[168]
XAV939	CyclinD1 and β-catenin	Enhances the expression of ADIPOQ and PLIN genes in leiomyoma cells, induced adipocyte trans-differentiation.	Leiomyoma	[169]
Bortezomib	β-catenin	Overexpression of α-catenin in adipocytes	Drosophila	[170]
E2F1	β-catenin through CTNNBIP1	Promotes differentiation and adipogenesis by activating ICAT in pre-adipocytes.	3T3-L1 preadipocytes, Hela, HEPG2	[171]
KYA1797K	β-catenin	Down-regulate the expression of STT3A/B, inhibit PD-L1 glycosylation, promote apoptosis in CSCs and inhibit immune evasion of CSCs.	Colon cancer	[172]
Actinomycin D and Telmisartan	β-catenin	Inhibits CSC expansion in tumours.	Lung Cancer	[173]
p53-Activator Wnt Inhibitor-2	β-catenin transactivation of downstream genes	Inhibits CSCs (i.e., hPCSCs, FGβ3 cells) growth.	Pancreatic cancer	[174]
PRI-724	CyclinD1	Reverses stemness caused by IR-MSCs.	Liver cancer	[175]
PKF118-310	β-catenin–TCF interaction	Inhibits CSCs growth.	Breast cancer	[176]
miR-454 of exosomes	PRRT2 (Proline-rich transmembrane protein2)	Inhibits stemness.	Ovarian cancer	[177]
G007-LK	Stabilises axin through TNKS1/2	Inhibits CSCs growth.	Glioma	[178]
XAV939 and aquaporin-3	β-catenin	Inhibits stemness.	Lung cancer	[179]
Poziotinib	Reduced wnt signalling	Inhibits stemness.	Ovarian cancer	[180]
Salinomycin	LRP6 co-receptor	Inhibits stemness.	Liver cancer	[181]
miR-601	Keratin5 (KRT5)	Promotes apoptosis.	Prostate cancer	[182]
DKK-1	Fzd or LRP5/6	Inhibits self-renewal, proliferation, migration, and tumorigenicity of CSCs.	Liver cancer	[183]
CGX1321	Wnt ligand synthesis	Eliminate CSCs.	HEK293 and Xenograft models	[184]
Salinomycin	β-catenin/T-cell factor complex	Reduced the expression of CSC-related Wnt target genes, including LGR5.	Colon cancer	[185]
Longdaysin	CK1δ and CK1ε; LRP6 and DVL2	Inhibits CSCs growth.	Breast cancer	[186]
Phenethyl isothiocyanate	β-catenin	Inhibits CSCs growth.	Colon cancer	[187]
Pyrvinium Pamoate	β-catenin	Inhibits CSCs growth.	Breast cancer	[188]
IC-2	β-catenin transcriptional activity	Inhibits and sensitises CSCs.	Colon cancer	[189]
IC-2	β-catenin transcriptional activity	Reduces the CD44-positive CSCs.	Liver cancer	[190]
HC-1	β-catenin signal	Reduces the CD44-positive CSCs.	Oral cancer	[191]
CWP232228	β-catenin-TCF binding	Inhibits the self-renewal capacity of CSCs.	Liver cancer	[192]
Evodiamine	β-catenin, c-Myc, and cyclin D1	Inhibits the self-renewal of CSCs.	Gastric cancer	[193]
sFRP4	Fzd receptor/Wnt lignads	Chemo-sensitization of CSCs.	Breast, Prostate, Ovary	[194]

**Table 2 cancers-15-05847-t002:** Exploring the interplay between Wnt signalling and the tumour microenvironment for targeted therapies.

Open-Ended Questions in the Interplay of Wnt Signalling and TME towards the Development of Advanced Therapeutics
How do neighbouring normal stromal cells transition to cancer-associated stromal cells? Does the Wnt signalling pathway activate or deactivate during this transition? What mechanisms allow tumour cells to evade immune system invasion? Are there specific markers indicating a transition to ECs, CAFs, TAMs, cancer-associated immune cells, dendritic cells, and cancer stem cells linked to Wnt signalling? Can biopsy-detectable biomolecules bridge TME and Wnt signalling in this transition? What strategies hinder cancer-associated stromal cell transformation clinically? How do we selectively target intercellular communication, non-coding RNAs, cancer stem cells, endothelial cells, and extracellular vesicles to counteract transition and drug resistance? Can Wnt-targeting agonists or antagonists prevent Wnt-driven TME evolution? How does single-cell RNA sequencing reveal the specific alterations induced by Wnt signalling in the TME during tumorigenesis, providing potential targets for therapeutic intervention? In Wnt-driven TME, how can CAR T-cell therapy be tailored based on single-cell RNA sequencing data to enhance its effectiveness in precisely targeting and disrupting Wnt-associated pathways in cancer cells?

## Data Availability

Data are contained within the article.

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
