# Peer review of "Navigating Tumour Microenvironment and Wnt Signalling Crosstalk: Implications for Advanced Cancer Therapeutics"

_cancers, 2023, doi:10.3390/cancers15245847_

Round 1
Reviewer 1 Report
Comments and Suggestions for Authors
Shraddha Shravani Peri et al. reviewed Crosstalk of Tumour Microenvironment and Wnt Signaling. This study is interesting and give insight into understanding of TME, and provide knowledge of cancer immunotherapy. I have some concerns:
1. Although this manuscript is well prepared, the language should be polished by a native speaker at deep extent.
2. The TME include soluble compartment (hormones, cytokines), immune cells and stroma cells, as well as extracellular matrix, while the author mainly introduce the role of WNT signaling in cells compartments, while its impact on cytokines should not be ignored, which play critical role in shaping TME.
3. Wnt signaling, and its crosstalk with immune cells, has both negative as well as positive effects on tumor progression. The author should summarize the clinical trials targeting aberrant Wnt signaling and the limitations of the strategies.
4. Wnt signaling affects cancer and immune surveillance, whether activation of Wnt signaling could results in increased sensitivity or resistance to immunotherapies is an interesting issue.
5. CAR T cell therapy is an innovative approach to treat cancers, some results demonstrated that the combination of CAR T-cell therapy with the Wnt inhibitor can increase the efficacy of CAR T cells in treating solid tumors, which should be summarized.
6. Some sections should be introduced more clearly, for example, the effect of Wnt signaling on DC, NK, MDSC, T cells and B cells should be separated. In addition, the mechanism should be introduced. It seems that the author only describe the results.
7. Some parts are too long and should be refined.
8. Single-cell RNA sequencing (scRNA-seq) could provide in-depth analysis of cell heterogeneity, which has also been used in the analysis of microenvironment, and there are researches about Wnt/β-catenin-mediated transcriptional change in single cells and their relationship with the microenviroment. Single-cell response to Wnt signaling activation should be briefly summarized in the reviewed.
9. Some classical literature about WNT signaling and tumor microenvironment should be cited. Some recent research study should be mentioned.
Comments on the Quality of English Languageminior editing needed
Author Response
Enclosure 2:
Author's Response to Reviewers' Comments: Navigating Tumour Microenvironment and Wnt Signalling Crosstalk: Implications for Advanced Cancer Therapeutics" (Manuscript ID:2713154)
We are grateful for the opportunity to revise our manuscript based on the insightful feedback from the reviewers. We have carefully addressed each point and implemented the following changes:
Reviewer #1:
1. Although this manuscript is well prepared, the language should be polished by a native speaker at deep extent.
We thank you for this suggestion. Professor Arun Dharmarajan, our co-corresponding author, boasting nearly four decades of experience across the US and Australia, has meticulously scrutinized the manuscript for clarity, time and again. Furthermore, Dr. Frank Arfuso, a native English speaker and co-author, has generously lent his expertise to elevate the readability. Notably, English has been the primary medium of education throughout the academic journeys of all co-authors, from their earliest schooling to their highest degrees. We've also collectively reviewed the manuscript, meticulously ensuring clarity and flow. We are confident the manuscript now gleams with conciseness and readability
2. The TME include soluble compartment (hormones, cytokines), immune cells and stroma cells, as well as extracellular matrix, while the author mainly introduce the role of WNT signalling in cells compartments, while its impact on cytokines should not be ignored, which play critical role in shaping TME.
Thank you for this suggestion. A separate paragraph on the role of cytokines and its impact on wnt signalling in the tumorigenesis have been included in Section 2 [lines 131-134] on Wnt-mediated Immune Editing with relevant citations.
3. Wnt signalling, and its crosstalk with immune cells, has both negative as well as positive effects on tumour progression. The author should summarize the clinical trials targeting aberrant Wnt signalling and the limitations of the strategies.
There are already two review articles existing summarizing the Wnt inhibitors targeting various components of Wnt signaling and immune checkpoint inhibitors that are under clinical trials. These articles are referenced in Section 9 [Consequence of Wnt inhibition on the Tumour microenvironmentLines 548-553].
4. Wnt signalling affects cancer and immune surveillance, whether activation of Wnt signalling could results in increased sensitivity or resistance to immunotherapies is an interesting issue.
Indeed the impact of Wnt signalling on regulation of response to immunotherapies is interesting. In section 2 on Wnt-mediated Immune Editing, we had discussed the impact of Wnt on immune surveillance. The biphasic effects of Wnt signalling especially non-canonical pathway activation on
immune cells and immunotherapy are discussed in Section 2 [lines 192-213] cited with relevant articles.
5. CAR T cell therapy is an innovative approach to treat cancers, some results demonstrated that the combination of CAR T-cell therapy with the Wnt inhibitor can increase the efficacy of CAR T cells in treating solid tumours, which should be summarized.
As per your suggestion, we have included a separate sub-section 2.1 [ Wnt signalling in Cancer Immunotherapies- Lines 226-256] describing the influence of Wnt signalling on immunotherapies. The paragraph also describes the significance of CAR T therapy and combination treatment modalities with
Wnt inhibitors cited with relevant articles. Emphasising the importance of this interaction, we have also discussed the how CAR T cell therapy could be strategized with Wnt inhibitors and is included in the last Section on Future Aspects [lines 585-588].
7. Some parts are too long and should be refined.
All the authors went through the manuscript for clarity and coherence. Sections that are long are refined for better understanding.
8. Single-cell RNA sequencing (scRNA-seq) could provide in-depth analysis of cell heterogeneity, which has also been used in the analysis of microenvironment, and there are researches about Wnt/β-catenin-mediated transcriptional change in single cells and their relationship with the microenviroment. Single-cell response to Wnt signalling activation should be briefly summarized in the reviewed.
The response of Wnt signalling and TME assessed through single-cell RNA sequencing are summarized in Section 2.1 [lines 165-172]. In addition, how sc-RNA seq technique could be used for future research is included in Section 10 [Future Aspects – lines 570-575].
9. Some classical literature about WNT signalling and tumour microenvironment should be cited. Some recent research study should be mentioned.
Recent and classical literature on Wnt signalling and TME have been cited in the text. The entire manuscript has been checked for recent and relevant citations.
Reviewer 2 Report
Comments and Suggestions for Authors
This manuscript deals with the relevance of wnt pathway in tumor microenvironment. The topic is of interest and the review is well organized and presented.
There are some concerns that the author should respond.
Some relevant papers on the topic are not considered and discussed. Please re.analyse the literature, insert and discuss the more recent advances on the topic.
Herein, I will list some of these works to be considered.
WNT signaling in the tumor microenvironment promotes immunosuppression in murine pancreatic cancer.
J Exp Med. 2023 Jan 2;220(1):e20220503. doi: 10.1084/jem.20220503. Epub 2022 Oct 14. PMID: 36239683
Cancer-cell-derived GABA promotes β-catenin-mediated tumour growth and immunosuppression
Wnt signaling arrests effector T cell differentiation and generates CD8+ memory stem cells. Nat Med. 2009 Jul;15(7):808-13. doi: 10.1038/nm.1982. Epub 2009 Jun 14.PMID: 19525962 Metastatic Latency and Immune Evasion through Autocrine Inhibition of WNT.Cell. 2016 Mar 24;165(1):45-60. doi: 10.1016/j.cell.2016.02.025.PMID: 27015306 Paradoxical cancer cell stimulation by IFNγ drives tumor hyperprogression upon checkpoint blockade immunotherapy. Cancer Cell. 2023 Feb 13;41(2):229-231. doi: 10.1016/j.ccell.2023.01.006.PMID: 36787694
The effect of Wnt/β-catenin signaling on PD-1/PDL-1 axis in HPV-related cervical cancer.
Oncol Res. 2023 Jan 12;30(3):99-116. doi: 10.32604/or.2022.026776. eCollection 2022.PMID: 37305016
Wnt activation promotes memory T cell polyfunctionality via epigenetic regulator PRMT1.
J Clin Invest. 2022 Jan 18;132(2):e140508. doi: 10.1172/JCI140508
Transforming dysfunctional CD8+ T cells into natural controller-like CD8+ T cells: can TCF-1 be the magic wand?
J Clin Invest. 2022 Jun 1;132(11):e160474. doi: 10.1172/JCI160474.PMID: 35642630
Highly immunogenic cancer cells require activation of the WNT pathway for immunological escape.
Sci Immunol. 2021 Nov 12;6(65):eabc6424. doi: 10.1126/sciimmunol.abc6424. Epub 2021 Nov 12.PMID: 34767457
Comments on the Quality of English LanguageEnglish language is good.
Author Response
Reviewer #2:
1. According to the World Health Organization’s Global Health Estimates, the primary cause of fatality worldwide in 2020 was ischemic heart disease (16%), followed by stroke (11%) and chronic obstructive pulmonary disease (6%) – please mind the fact these statistics were reported pre-COVID pandemic and should be updated soon. Although cancer has a high mortality rate, it ranks 6th among the causes of mortality. Please check the first line of the introduction for accuracy.
2. Cancer is characterised by diverse mechanisms besides “uncontrolled growth of cells without normal boundaries.” – please check the latest update on the Hallmarks of Cancer (doi: 10.1158/2159-8290.CD-21-1059) for a more accurate definition.
As suggested in the point 1 and 2, the statistics and definition of Cancer was verified and modified. In Section 1 [Introduction –lines 46-50], cancer is defined after referring the recent hallmarks of cancer and the relevant article has been cited to provide accuracy to the content.
3. Line 112: Formatting needed on Ca2+ - superscript.
Line 112, marked in the original draft, is the description of Figure 2. As recommended, the Ca2+ was formatted with superscript [Ca2+] lines position changed to 108.
4. The paragraph from lines 156 and 189 should be segmented to give significant meaning to each topic discussed. We suggest the authors transform the paragraph into two segments, ranging from line 156 to line 174 (discussing TILs and their inhibition), and then from line 174 to line 189 (discussion over immune inhibition of the Wnt components).
As recommended, Section 2 [Wnt-mediated immune editing- Lines 173-191] discussing TILs and inhibition ; lines 192-213 discussing immune inhibition by Wnt components] is segmented into two paragraphs and revised for better clarity. The position of the lines changed during revision.
5. Figures 3, 4, and 5 need to be referenced in the text before their appearance in the manuscript. Figure legends need to be broadened more.
All the figures are referenced inside the text as recommended. Figure 1 and 2 is referenced at the end of the 2nd paragraph in Section 1 [Introduction]. Figure 3 is referenced in the 3rd paragraph in Section 1 [Introduction]. Figure 4 and 5 is referenced in Section 9 [Consequence of Wnt inhibition on the Tumour microenvironment]. All the legends of the Figures are elaborated for better
understanding.
6. Please note that the use of the term “exosomes” is not recommended by the International Society of Extracellular Vesicles (MISEV2018, doi: 10.1080/20013078.2018.1535750) since it implies vesicles with an endosomal biogenesis pathway whose size and density overlaps other extracellular particles. Although the cited paper has used the term exosomes, we suggest replacing the term with the more correct “extracellular vesicles”.
Thank you for that valuable information. As suggested in Section 3, the word ‘exosomes’ was replaced by ‘extracellular vesicles.’ The paragraph was read for better clarity.
7. The paragraph from lines 245 to 253 lacks references – please revise and report sentences with the appropriate references.
The paragraph lines mentioned by the author fall on Section 4 [titled: Wnt signalling in tumour adipocytes: Regulation and impact – Lines 300-308]. As suggested, the texts were cited with four relevant articles, and the paragraph was reworked for better understanding.
8. Paragraphs ranging from lines 268 to 274 and 275 to 280 should be joined since they discuss the same topic.
The paragraph lines mentioned by the author to be joined fall on Section 4 [titled: Wnt signalling in tumour adipocytes: Regulation and impact Lines 323-334]. As suggested, the paragraphs were joined and revised for better clarity and coherence. The position of the lines changed during revision.
9. Tumour vasculature is not well-organized (line 282) as described by the authors. It has been reported as abnormal, leaky, and prone to structural damage due to their lack of pericytes to sustain the vessels (doi: 10.1038/nrc724; doi: 10.1038/s41419-017-0061-0). Please revise the paragraph.
Thank you for pointing this out. As suggested by the reviewer, tumour vasculature has been introduced with the recommended article cited in Section 5 [tilted: Wnt signalling mediated modelling of tumour vasculature and lymphangiogenesis – Lines 336-344]. The paragraph was revised for better coherence.
10. Figure 4: The text inside of the figure needs to be adjusted not to overlap the images (i.e., on the endothelial cells section, the text is overlapped by the vessel). We also suggest increasing the font size of the smaller text for better visibility.
Figure 4 was reformatted again by increasing the font size and positioning of the boxes and texts, and it was checked for overlaps.
11. Formatting is needed on lines 53 and 467 (extra space).
As suggested, the text in lines 56 and 535 [position changed during , which were marked in the draft at submission, was formatted for line and paragraph spacing.
12. Table 1 needs to be referenced in the text. Table 1 is referenced in Section 9 [tilted: Consequence of Wnt inhibition on the Tumour microenvironment- Line 546].
13. Comments on the Quality of English Language: The quality of English was acceptable; however, the text needs to undergo revision for better clarity and pace. Sentences were often disconnected from each other within the same paragraph and could be rearranged to improve the reader´s understanding of the science behind them.
The authors have now revised the draft for better coherence.

Reviewer 3 Report
Comments and Suggestions for Authors
In this review, the authors dissect the multiple potential applications of the Wnt signaling pathway in the treatment of cancer by targeting the tumor microenvironment. They describe in detail the evidence behind the diverse cell types regulating TME, including immune cells, cancer stem cells, and endothelial cells, and the importance of Wnt for the development of cancer-like behaviors in cells of the TME. Overall, the writing was enjoyable and clear, although some paragraphs should be revised for clarity and some basic concepts need to be reassessed. There is room for improvement in the figures, which should be remade for better resolution, clarity and understanding - particularly, text should be visible and images, when shown, should be adjusted to not overlap text.
Point-by-point suggestions:
-
According to the World Health Organization’s Global Health Estimates, the primary cause of fatality worldwide in 2020 was ischemic heart disease (16%), followed by stroke (11%) and chronic obstructive pulmonary disease (6%) – please mind the fact these statistics were reported pre-COVID pandemic and should be updated soon. Although cancer has a high mortality rate, it ranks 6th among the causes of mortality. Please check the first line of the introduction for accuracy.
-
Cancer is characterized by diverse mechanisms besides “uncontrolled growth of cells without normal boundaries” – please check the latest update on the Hallmarks of Cancer (doi: 10.1158/2159-8290.CD-21-1059) for a more accurate definition.
-
Line 112: Formatting needed on Ca2+ - superscript.
-
The paragraph from lines 156 and 189 should be segmented to give significant meaning to each topic discussed. We suggest the authors transform the paragraph into two segments, ranging from line 156 to line 174 (discussing TILs and their inhibition), and then from line 174 to line 189 (discussion over immune inhibition of the Wnt components).
-
Figures 3, 4, and 5 need to be referenced in the text before their appearance in the manuscript. Figure legends need to be broadened more.
-
Please note that the use of the term “exosomes” is not recommended by the International Society of Extracellular Vesicles (MISEV2018, doi: 10.1080/20013078.2018.1535750) since it implies vesicles with an endosomal biogenesis pathway whose size and density overlaps other extracellular particles. Although the cited paper has used the term exosomes, we suggest replacing the term with the more correct “extracellular vesicles”.
-
The paragraph from lines 245 to 253 lacks references – please revise and report sentences with the appropriate references.
-
Paragraphs ranging from lines 268 to 274 and 275 to 280 should be joined since they discuss the same topic.
-
Tumor vasculature is not well-organized (line 282) as described by the authors. It has been reported as abnormal, leaky, and prone to structural damage due to their lack of pericytes to sustain the vessels (doi: 10.1038/nrc724; doi: 10.1038/s41419-017-0061-0). Please revise the paragraph.
-
Figure 4: The text inside of the figure needs to be adjusted not to overlap the images (i.e., on the endothelial cells section, the text is overlapped by the vessel). We also suggest increasing the font size of the smaller text for better visibility.
-
Formatting is needed on lines 53 and 467 (extra space).
-
Table 1 needs to be referenced in the text.
The quality of English was acceptable, however the text needs to undergo revision for better clarity and pace. Sentences were often disconected from each other within the same paragraph and could be rearranged to improve the reader´s understanding of the science behind.
Author Response
Reviewer #3:
1. Some relevant papers on the topic are not considered and discussed. Please reanalyse the literature insert and discuss the more recent advances on the topic.
As suggested by the reviewer, all nine articles were inserted inside the relevant topic, and the literature was reanalysed for better coherence.
Round 2
Reviewer 2 Report
Comments and Suggestions for Authors
The Authors have improved the manuscript.
Comments on the Quality of English LanguageEnglish is good enough.
Reviewer 3 Report
Comments and Suggestions for Authors
The revisions improved the paper's quality and it is now fit for publishing.